# Metabolomics Combined with Multivariate Statistical Analysis for Screening of Chemical Markers between *Gentiana scabra* and *Gentiana rigescens*

**DOI:** 10.3390/molecules25051228

**Published:** 2020-03-09

**Authors:** Gaole Zhang, Yun Li, Wenlong Wei, Jiayuan Li, Haoju Li, Yong Huang, De-an Guo

**Affiliations:** 1School of Pharmaceutical Sciences, Changchun University of Chinese Medicine, Changchun 130117, China; nancyzgl@163.com; 2Shanghai Research Center for Modernization of Traditional Chinese Medicine, National Engineering Laboratory for TCM Standardization Technology, Shanghai Institute of Materia Medica, Chinese Academy of Sciences, Haike Road 501, Shanghai 201203, China; liyunsci@163.com (Y.L.); 13521032532@163.com (W.W.); li_jiayuan@simm.ac.cn (J.L.); lihaojv18@mails.ucas.ac.cn (H.L.); huang_yong@simm.ac.cn (Y.H.); 3Department of Traditional Chinese Medicine, China Pharmaceutical University, Tongjiaxiang 24, Nanjing 210009, China

**Keywords:** Gentianae Radix et Rhizome, *Gentiana scabra*, *Gentiana rigescens*, metabolomics, UPLC/LTQ-Orbitrap-MS, chemical markers

## Abstract

Gentianae Radix et Rhizome (Longdan in Chinese, GRR) in Chinese Pharmacopoeia is derived from the dried roots and rhizomes of *Gentiana scabra* and *G. rigescens*, that have long been used for heat-clearing and damp-drying in the medicinal history of China. However, the characterization of the chemical components of two species and the screening of chemical markers still remain unsolved. In current research, the identification and characterization of chemical components of two species was performed using ultra-high-performance liquid chromatography (UHPLC) coupled with linear ion trap-Orbitrap (LTQ-Orbitrap) mass spectrometry. Subsequently, the chemical markers of two species were screened based on metabolomics and multivariate statistical analysis. In total, 87 chemical constituents were characterized in *G. scabra* (65 chemical constituents) and *G. rigescens* (51 chemical constituents), with 29 common chemical constituents being discovered. Thereafter, 11 differential characteristic components which could differentiate the two species were designated with orthogonal partial least squares discriminant analysis (OPLS-DA) and random forest (RF) iterative modeling. Finally, seven characteristic components identified as (+)-syringaresinol, lutonarin, trifloroside, 4-*O*-β-d-glu-trifloroside, 4″-*O*-β-d-glucopyranosy1-6′-*O*-(4-*O*-β-d-glucaffeoyl)-linearroside, macrophylloside a and scabraside were selected as the chemical markers for the recognition of two *Gentiana* species. It was implied that the results could distinguish the GRR derived from different botanical sources, and also be beneficial in the rational clinical use of GRR.

## 1. Introduction

Gentianae Radix et Rhizoma (GRR), named Longdan in Chinese, is a widely used as traditional Chinese medicine (TCM) for the treatment of icterepatitis, dermatophytes and herpes zoster [1]. As first recorded in *Shennong Bencao Jing* (*Shennong’s Classic of Materia Medica*) in Han Dynasty, GRR has been used as hepatoprotective and choleretic drug for thousands of years. It has also been described in various Chinese ancient medicinal monographs, such as *Tujing Bencao* in Song Dynasty and *Bencao Gangmu* in Ming Dynasty. It was mainly used for clearing liver and gallbladder dampness and heat, purging fire of the liver and gallbladder, relaxing tendons and relieving pain, etc. [2,3,4]. Modern pharmacological studies reported that it possessed anti-inflammatory, anti-oxidative and antiviral activities [5,6,7,8]. According to phytochemical research, the main constituents of GRR are iridoids, xanthones, flavonoids, triterpenoids, and others [4,9,10,11]. Arrays of studies demonstrated that iridoids and xanthones were the active components and had various remarkable pharmacological activities including hepatoprotective [12], anti-inflammatory [13,14,15], antioxidant [8,16,17] and immunomodulatory activities [17]. However, investigations into the chemical profiling of GRR and screening of chemical markers of GRR derived from the different sources remain inadequate.

It is well known that morphologic and microscopic identifications, high-performance thin-layer chromatography (HPTLC) technique, DNA barcodes and metabolomics could all be used for the interspecies identification [18]. However, the there is a high possibility of false identification by applying morphologic and microscopic techniques, due to the subjective judgement by the experimenters. Although the TLC and DNA barcoding techniques were more objective and accurate compared to other approaches, the results lacked integrated adequate chemical information, and could not screen chemical markers to identify and classify different species in an efficient and swift manner. Metabolomics revealed the global chemical profiling of TCM and the raw data could be further analyzed with multivariate statistical analysis for the screening of chemical markers among different species or sources [19].

Nowadays, ultra-high-performance liquid chromatography (UHPLC) coupled with high-resolution mass spectrometry (HRMS) has been widely applied for the chemical profiling of complex mixtures from natural products [20,21]. LC-HRMS could offer the potential chemical composition and infer the detailed structures of analytes on the basis of the measurement of accurate mass and generation of the MS/MS or MS^n^ data [22,23,24]. Moreover, it could determine hundreds or even thousands of MS features using a single injection in a short time and with less consumption of organic solvents, and implement high-throughput data acquisition [25].

In this study, an untargeted metabolomics combined with univariate analysis, orthogonal partial least-squares-discriminant analysis (OPLS-DA), and random forest (RF) for the screening of chemical markers which could be applied to accurately distinguish *G. scabra* and *G. rigescens*. Furthermore, the characterization of chemical components of GRR was performed by UHPLC-LTQ-Orbitrap/MS. The common and specific chemical constituents of *G. scabra* and *G. rigescens* were compared and classified accordingly.

## 2. Result and Discussion

### 2.1. Optimization of Chromatographic Conditions and Sample Extraction

In order to obtain good separation, the different chromatographic columns, gradient elution program, column temperature and flow rate were optimized in detail. The six different columns including Eclipse Plus C18, Extent C18, BEH C18, Kinetex C18, Zorbax SB C18 and HSS T3 were used to compare the separation capacity, the detailed column information was listed in Appendix A. As a result, the HSS T3 column showed a better retention capacity and column efficiency for the separation than the other columns. Subsequently, the flow rate (0.3, 0.4, and 0.5 mL/min) and column temperature (25, 30, 35, and 40 ℃) were compared under the optimized gradient elution program, and 0.4 mL/min at 35 ℃ was selected for separation (Figure 1). The injection volume was 2 μL with no solvent effect. In addition, QC samples of GRR were collected in both positive and negative ion modes, and more useful information was prevailed in negative mode (Figure 2).

Different extract solvents (methanol, ethanol and acetonitrile) were compared according to the response intensity of four standards (longanic acid, gentiopicroside, sweroside and trifloroside), and the result of the extraction of methanol solvent was better than others. The solvent ratio was evaluated as above, and the extraction efficiency of 50% methanol–water (*v/v*) was optimal (Figure 3).

### 2.2. Characterization of Chemical Components by UPLC-LTQ-Orbitrap/MS

#### 2.2.1. Deductive Fragmentation Pathway of Iridoids

Iridoids belong to the highly oxygenated monoterpene; its parent nucleus was a five-carbon cyclopenta pyranoid skeletal structure. In the simple iridoids, C-1 semi-acetal hydroxyl was active, and always linked with β-d-glucose to form glycosides, mostly mono-glycosides.

Some iridoids, known as the seco-iridoids, such as 7,8-secoderivative, were formed by the cleavage of cyclopentane ring at the bond of C-7 and C-8. In the seco-iridoids, it could be calssified into four categories: gentiopicrins, swerosides, swertiamarinss and dicarboxylic acids. The structure of gentiopicrin was characterized by the formation of double bonds between C-3 and C-4 and C5 and C-6 positions, based on the seco-iridoid parent nucleus. To differentiate from gentiopicrin, sweroside glycoside’s skeletal structure only has one double bond in C-3 and C-4. Swertiamarin glycoside has the hydroxyl substituents at C-5 on the basis of the sweroside parent nucleus structure. Both sweroside and swertiamarin had a glycosyl–acetylation structure. The structure of rindoside, one compound of swertiamarin group, was taken as an example for summarizing the cleavage pattern: de-saccharification occurs first and then deacetyl group as shown in Figure 4 [26].

#### 2.2.2. Deductive Fragmentation Pathway of Flavonoids 

Due to the stable conjugated structure of flavonoids, cleavage often occurred on the sugar moiety attached to the A ring, such as homoorientin shown in Figure 5. The homoorientin generated its [M-H]^−^ ion at *m/z* 447.15; it produced [M-H-H_2_O]^−^ ion at *m/z* 430.09, or generated a series of ion at *m/z* 358.06 [M-H-C_3_H_6_O_3_]^−^, *m/z* 328.05[M-H-C_3_H_6_O_3_-OH]^−^and *m/z* 299.06 [M-H-C_3_H_6_O_3_-OH-CH_2_O]^−^. The proposed fragmentation pathway was similar with deductive homoorientin, therefore, the other 13 flavonoid compounds were tentatively characterized [27].

#### 2.2.3. Deductive Fragmentation Pathway of Xanthones 

The fragment ion formation process of mangiferin and mechanism of mass spectrometry fragmentation are shown in Figure 6. Mangiferin displayed [M-H]^−^ ion at *m/z* 421.08, then generated two kinds of ion: *m/z* 259.02 [M-H-Glu]^−^ or *m/z* 332.05 [M-H-C_3_H_6_O_3_]^−^, and the major ion *m/z* 332.05 [M-H-C_3_H_6_O_3_]^−^ successively produced *m/z* 302.04 [M-H-C_3_H_6_O_3_-CH_2_O]^−^ and 272.03 [M-H-C_3_H_6_O_3_-CH_2_O]-[28].

According to the cleavage regularity and detected chemical data information, in total, 87 compounds, including 54 iridoids, 13 flavonoids, two xanthone, four triterpenoids and five other components were characterized in *G. scabra* (65 chemical constituents) and *Gentiana rigescens* (51 chemical constituents), among which 29 common chemical constituents were shown in Table 1 and Figure 7.

### 2.3. Screening of Chemical Markers by Metabolomics and Chemometrics

#### 2.3.1. Univariate Analysis for the Screening of Differential Metabolites

Compared with multivariate statistics, univariate analysis focuses more on independent changes in the levels of metabolites. After the data filtering with 80% and 15% rules, the 366 stable metabolic characteristics were obtained. Student’s *t*-test was applied to figure out the features which had statistical difference (*p*-value < 0.05) between the two groups. The results showed that 283 features with *p*-value < 0.05 were screened out, and the heat map (Figure 8) illustrated the differences in 283 features between the two *Gentiana* species. It can be seen clearly that the 283 screened features accumulated in the two *Gentiana* species show great differences; about half of the metabolites were upregulated in the *G. scabra* group compared with their counterparts in the *G. rigescens* group.

#### 2.3.2. Data Visualization and Experimental Stability Evaluation

Principal component analysis (PCA) was carried out to provide an unbiased visual representation of the sample distributions in the extracted principal components (PCs) space. As a result, after the Pareto scaling, the first seven PCs of the PCA-X model explained 90.6% of the variation in the original dataset (R^2^X(cum) = 0.906); 74.4% of the variation in original data was predicted by the model according to 7-fold cross validation (Q^2^(cum) = 0.744), which means that this PCA-X model could well represent the variation information of the original data (Figure 9a). The sample distribution of two *Gentiana* species and QC samples in the score plot (PC_1_ vs. PC_2_) was provided in Figure 9b. The sample distribution of QC samples was closely clustered, showing that the data collection was relatively stable during the entire experiment. These test samples from different species are well separated, indicating that the differences in metabolic characteristics between two *Gentiana* species were obvious. Therefore, for the next step, the decision combination of OPLS-DA and RF were further used in search of the chemical markers which could be used to successfully distinguish two *Gentiana* species.

#### 2.3.3. Screening of Differential Metabolic Characteristics 

A supervised OPLS-DA approach was used to preliminarily investigate the metabolite characteristics that showed the greatest differences between the two *Gentiana* species. As a result, after the Pareto scaling, the OPLS-DA model described 86.5% of the variation in X (R^2^X (cum) = 0.865); 98.9% in response Y (R^2^Y(cum) = 0.989) and 97.5% in response Y was predicted by the model according to 7-fold cross validation (Q^2^(cum) = 0.975), with one predictive and six orthogonal (1 + 6) components. The high value of those parameters demonstrated that the OPLS-DA model presented a good classification and prediction ability to distinguish two class. In Figure 10a, the sample distribution in the score plot (predictive component 1 vs. X side orthogonal component 1) of two *Gentiana* species were well separated, and the interclass samples heterogeneity of *G. scabra* were larger than *G. rigescens*, which indicated that the chemical variation in different batches of *G. scabra* was larger than that of *G. rigescens*. Subsequently, permutation tests (n = 200) were performed to validate the model performance. As shown in Figure 10b, the values of R^2^ = (0.0, 0.394) and Q^2^ = (0.0, −0.664) of category 1 and R^2^ = (0.0, 0.385) and Q^2^ = (0.0, −0.672) of category 2 indicated the OPLS-DA model in the present study having no risk of overfitting. In total, 47 variables (VIP > 1, Figure 10c) in the OPLS-DA model were selected as the differential metabolic characteristics, since these variables have an important identification capability, which is also consistent with the result shown in S-plot (Figure 10d). These differential metabolic characteristics were helpful to clarify the chemical differences in two *Gentiana* species.

Although 47 differential metabolic characteristics selected by OPLS-DA could successfully identify two *Gentiana* species, many metabolic characteristics were not easy to detect in practice. Therefore, these differential metabolic characteristics need to be further refined to select a set of robust features labeled as chemical markers after characterization. Furthermore, the model performance has been greatly affected by the data scaling method [29], which may interfere with the correct selection of important variables in the metabolomic research. By contrast, RF was not very sensitive to the choice of data scaling method [30], and the RF model was further adopted, with 47 differential metabolic characteristics. The parameters of RF were set for 500 trees, and the *m*_try_ was the default value (square root of the number of variables). RF was modeled through 100 iterations; the variable importance of each modeling process was ranked, then the ranked variable ID was stored for further analysis.

In Figure 11a, according to the sorted table of the variable importance of 100 RF model iterations, the cumulative number of each variable in top Nth was obtained. Each line represented a metabolic characteristic; it could be clearly seen that the variables were divided into two parts. The cumulative number of 36 variables in the left side reached 100 around top 35th, while the remaining 11 variables were situated in the right side due to their relatively low feature importance. Then, 36 variables which showed a relatively larger feature importance in 100 iteration modeling were selected. Subsequently, the frequency of the variable as the most important parameter in the 100 iterative modeling was calculated. In Figure 11b, the variable with the highest frequency appeared nine times, indicating that this metabolic characteristic was of major importance in differentiating between the two *Gentiana* species. Furthermore, one variable appeared six times, four variables five times and five variables four times, and showed great discriminating power. In the results, 11 metabolic characteristics that appeared more than four times were selected. Finally, based on 11 selected metabolic characteristics, RF model (500 trees) was established to distinguish the two *Gentiana* species. Results showed that the out-of-bag estimate of error rate of RF model is 0, and the area under the curve (AUC) of receiver operating characteristic (ROC) curve of the RF model was 1.00, indicating that the two *Gentiana* species could be correctly identified using 11 selected metabolic characteristics.

#### 2.3.4. Identification of Chemical Markers

The 11 characteristic markers were putatively identified according to the retention time, accurate mass, MS^2^ and fragmentation pattern of standards. Finally, the seven markers were identified as (+)syringaresinol, lutonarin, trifloroside,4-β-d-glu-trifloroside, 4″-*O*-β-d-glucopyranosy1-6′-*O*-(4-*O*-β-d-glucaffeoyl) linearroside, macrophylloside A and scabraside respectively, and four markers remained unidentified (Table 2).

## 3. Experimental

### 3.1. Materials and Reagents

The standard compounds, gentiopicroside, swertiamarin, sweroside, loganin, homoorientin, mangiferin, were purchased from Shanghai Nature-Standard Biotech Co., Ltd (Shanghai, China); 6’-*O*-β-d-gentiopicroside from Shanghai Yuanye Bio-Technology Co., Ltd (Shanghai, China); secologanoside from Chengdu Priifa Technology Development Co. Ltd (Chengdu, China); loganic acid, trifloroside, rindoside and 6’-*O*-β-d-glu-loganic acid from Chengdu DeSiTe Biological Technology Co., Ltd (Chengdu, China).

HPLC-grade acetonitrile and formic acid used in the mobile phase were purchased from Merck KGaA (Darmstadt, Germany) and Tedia Company Inc. (Fairfield., OH, USA), respectively. Ultrapure water (18.2 MΩ cm at 25 °C) was in-lab prepared by a Millipore Alpha-Q water purification system (Millipore, Bedford, MA, USA). HPLC-grade methanol for the sample preparation was purchased from Sino Pharm Chemical Reagent Co., Ltd. (Shanghai, China).

### 3.2. Sample and Standards Preparation

The roots and rhizome of GS and GR were collected from Liaoning Province in September and Yunnan Province in China in November, 2019. Fifty-four batches of GS samples were collected from six areas of Qingyuan, particularly in Liaoning Province (the largest base for the cultivation of GS in China.) Thirty-four batches of GR samples were collected from four areas of Yunxian in Yunnan province. The collection information was summarized in Table 3. These samples were all recorded according to their resources. All of them were identified by Professor Jinglong Zhang, who is the expert in medicinal botany in Changchun University of Chinese Medicine. The plant specimens were stored in Shanghai Research Center for Modernization of Traditional Chinese Medicine, National Engineering Laboratory for TCM Standardization Technology, Shanghai Institute of Materia Medica, Chinese Academy of Sciences.

An aliquot of 0.250 g of accurately weighted fine powder of GRR was initially immersed in 10 mL 50% aqueous methanol (*v/v*) and extracted on a water bath at 40 °C with ultrasound (1130 W, 37 kHz) assistance for 30 min. The supernatant was transferred into a 2 mL centrifuge tube. After being centrifuged at 14,000 rpm in 10 min, the supernatant was filtered through a 0.22 μm PTFE microporous membrane (Agilent Technologies, Santa Clara, CA, USA) to prepare the test solutions. The standard solutions of gentiopicroside, swertiamarin, sweroside, loganin, homoorientin, mangiferin, 6’-*O*-β-d-gentiopicroside, secologanoside, loganic acid, trifloroside, rindoside and 6’-*O*-β-d-glu-loganic acid were prepared in methanol at the appropriate concentration.

### 3.3. Instrumentation and Condition

Chromatographic separation was executed on an Ultimate 3000 UHPLC system (Thermo Fisher Scientific, San Jose, CA, USA). The separation was performed on a Waters (Milford, MA, USA) HSS T3 column (2.1×100 mm, 1.7 μm) and maintained at 30 °C. The GRR samples were eluted with a mobile phase consisting of 0.1% formic acid water (A) and acetonitrile (B) in the following gradient program: 0–4 min, 10–20% B; 4–12 min, 12–27% B; 12–30 min, 27–53% B; 30–31 min, 53–100% B; 31–25 min, 100% B. The flow rate was set at 0.4 mL/min. The injection volume was 2 μL.

An LTQ-Orbitrap Velos Pro hybrid mass spectrometer (Thermo Fisher Scientific, San Jose, CA, USA) was used for accurate mass measurements and data collection in negative mode, and with ESI-source-operated. The ESI source parameters were set as follows: ion spray voltage 2.7 kV, capillary temperature 320 °C, source heater temperature 200 °C, sheath gas (N_2_) 15 arbitrary units, auxiliary gas (N_2_) eight arbitrary units, and sweep gas (N_2_) two arbitrary units. The Orbitrap analyzer scanned the mass range from *m/z* 50 to 1345 with a resolution of 30,000 (FWHM defined at *m/z* 400) for MS. The MS data were recorded in profile and centroid formats, respectively. The average acquisition time required to finish a scan circle (containing four scan events) was 1.8 s. Default values were used for most other acquisition parameters.

### 3.4. Data Processing and Multivariate Analysis

Progenesis QI (Waters, Milford, USA) was used to process the raw data acquired from the UPLC-LTQ-Orbitrap/MS. The feature detection, precursor ions fusion, and retention time correction operated by Progenesis QI to get a peak table (5528) including retention time, *m/z*, and normalized abundance of all samples. Regarding the data filtering, firstly, features with a relative standard deviation (RSD) greater than 15% in QC samples were removed because these features were unstable during the data collection of the entire experiment. Secondly, the remaining features were filtered according to the 80% rule, whereas features present in at least 80% of samples in one group were allowed to remain.

In data analysis, univariate and multivariate statistical methods were both introduced to contribute the complementary advantages [31]. Firstly, univariate data analysis, i.e., Student’s *t*-test, was used to figure out the features which had statistical difference (the *p*-value of a Student’s *t*-test of < 0.05) between the two groups. Secondly, a visual representation of the sample distributions for two *Gentiana* species and QC samples were provided by principal component analysis (PCA). Thirdly, OPLS-DA analysis was carried out to distinguish two *Gentiana* species, and the metabolic characteristics with VIP > 1 were chosen as differential metabolic characteristics. Finally, the RF analysis was iteratively modeled 100 times to deeply investigate the permutation accuracy importance of these differential metabolic characteristics, and the chemical markers were selected to differentiate two *Gentiana* species.

### 3.5. Software

The raw data of UPLC-LTQ-Orbitrap-MS were processed by Progenesis QI. Univariate analysis of Student’s *t*-test was calculated by Excel, heat map was formed by heatmap package (version 4.6-14) in an R environment (version 3.4.3). Multivariate analysis of PCA and OPLS-DA was processed by software SIMCA-P^+^ 13.0 (Umetrics AB, Umeå, Sweden). An RF model was established by random Forest package (version 4.6-14) in R environment (version 3.4.3).

## 4. Conclusions

A systematic chemical characterization method was developed to provide an effective and scientific basis for the quality control of GRR, as well as to carry out an integrated platform based on plant metabolomics and chemometrics for the characterization and classification of two officinal *Gentiana* species. In the present study, 87 components were identified in GRR by UHPLC-LTQ-Orbitrap/MS, including 54 iridoids, 13 flavonoids, two xanthone, and four triterpenoids. Then, after the data visualization and classification analysis of whole metabolite profiles operated by PCA and OPLS-DA, two *Gentiana* species of GRR were clearly separated and correctly identified, and 47 differential metabolic characteristics were selected. Subsequently, 11 differential features were further selected based on the 100 RF iterative modeling. After matching with authentic compounds, seven differential components were selected as chemical markers for the recognition of two *Gentiana* species. The results indicated that metabolomics combined with chemometrics is a powerful tool to differentiate closely related species and could be used as essential data for quality control of traditional Chinese medicines.

## Figures and Tables

**Figure 1 molecules-25-01228-f001:**
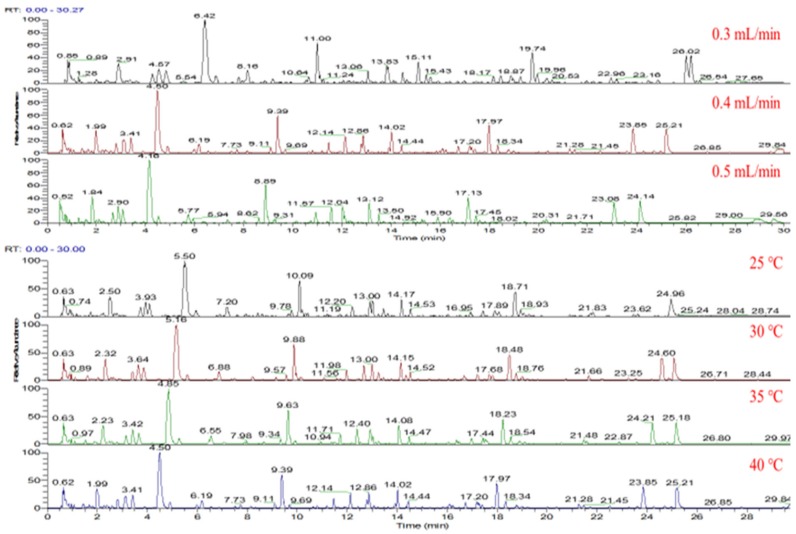
The typical Base Peak ion (BPI) of Gentianae Radix et Rhizome (GRR) chromatographic separation.

**Figure 2 molecules-25-01228-f002:**
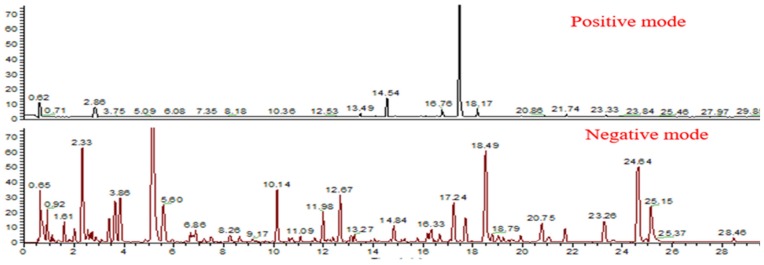
The typical BPI of GRR in positive and negative ion mode.

**Figure 3 molecules-25-01228-f003:**
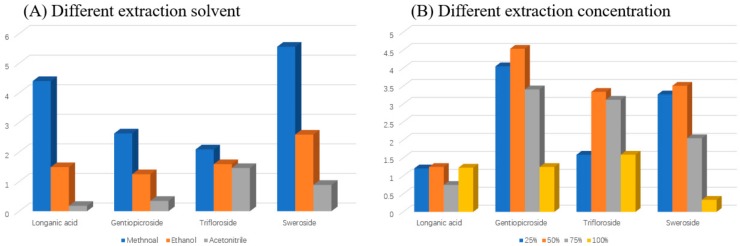
Optimization of sample extraction conditions: (**A**) Different extraction solvent, (**B**) Different methanol ratio.

**Figure 4 molecules-25-01228-f004:**
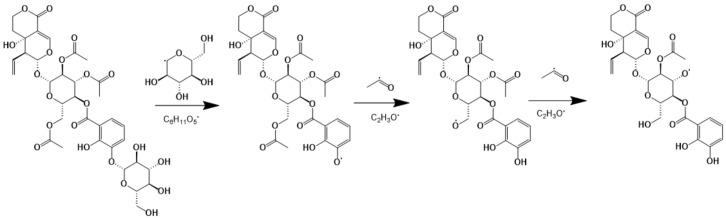
Deductive fragmentation pathway of rindoside.

**Figure 5 molecules-25-01228-f005:**
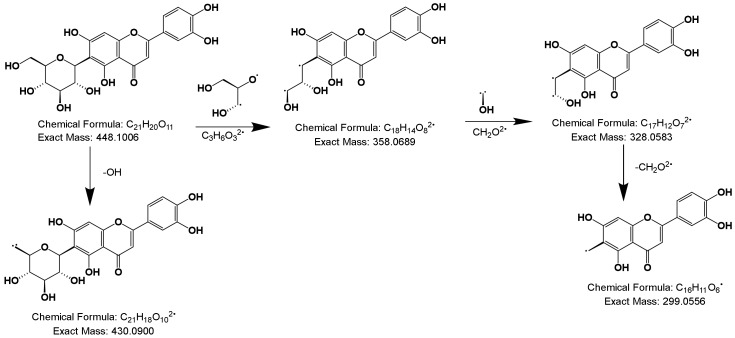
Deductive fragmentation pathway of homoorientin.

**Figure 6 molecules-25-01228-f006:**
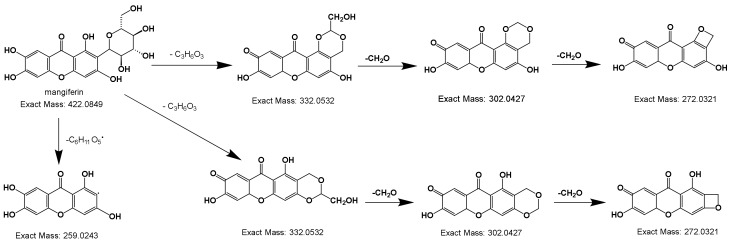
Deductive fragmentation pathway of mangiferin.

**Figure 7 molecules-25-01228-f007:**
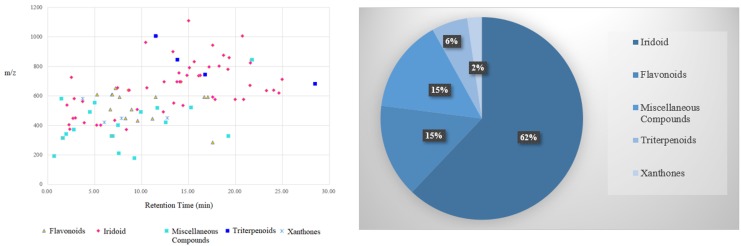
Characterization of chemical constituents of two species of GRR by UHPLC -LTQ-Orbitrap/MS in negative ion mode.

**Figure 8 molecules-25-01228-f008:**
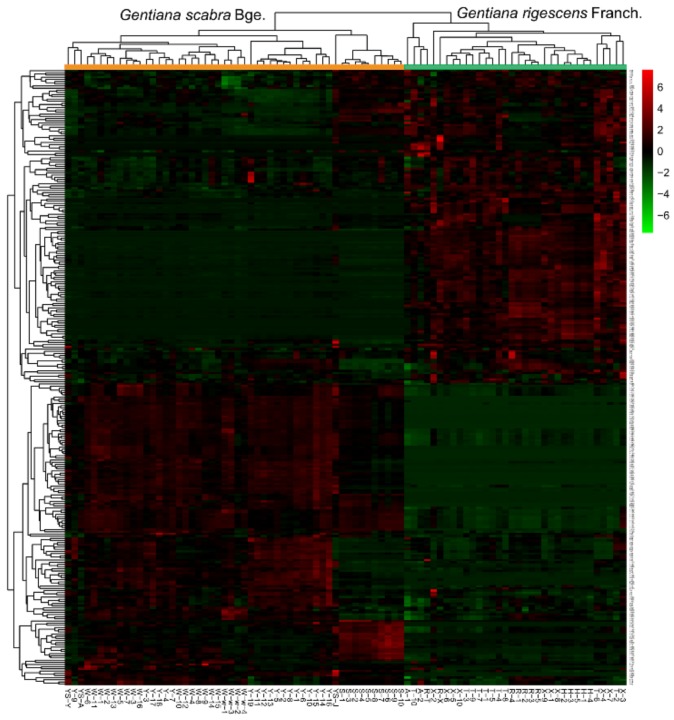
Heat map formed by 283 features (*p*-value < 0.05) in *Gentiana scabra* and *Gentiana rigescens.*

**Figure 9 molecules-25-01228-f009:**
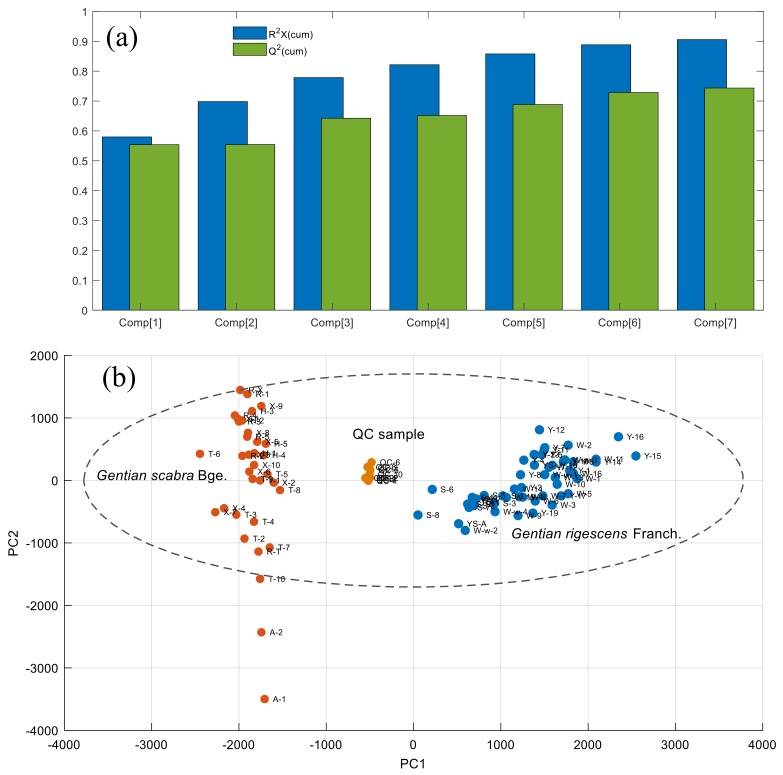
(**a**) Accumulated R2X and Q2 with different numbers of PCs. (**b**) score plot of PCA-X model for data visualization and experimental stability evaluation of two Gentiana species and QC samples.

**Figure 10 molecules-25-01228-f010:**
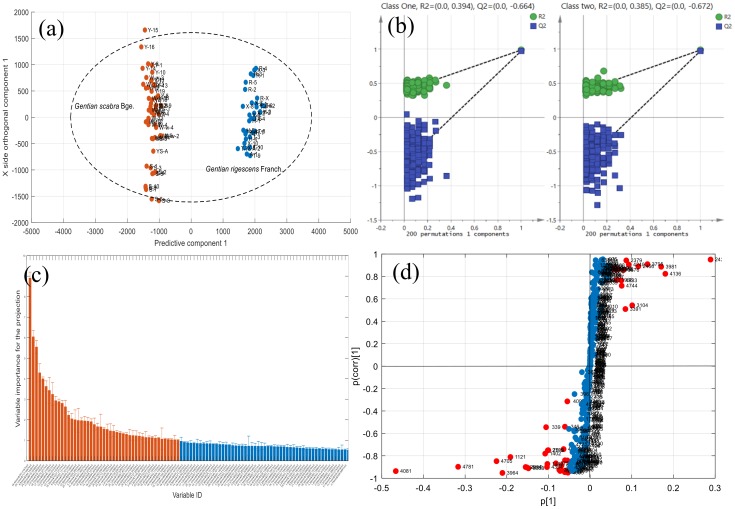
(**a**) Score plot of OPLS-DA model for differentiating two *Gentiana* species. (**b**) Permutation plot of two classes at 200 times of permutations. (**c**) Variable importance for the projection (VIP) of each variable. (**d**) S-plot of each variable.

**Figure 11 molecules-25-01228-f011:**
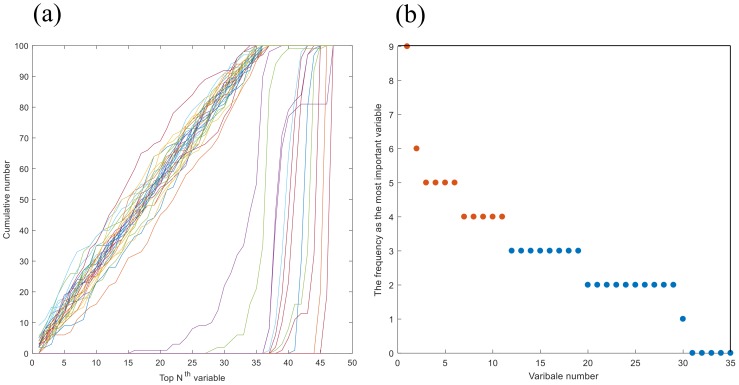
(**a**) Cumulative number of each variable in the top Nth of 100 RF iterative modeling. (**b**) Frequency of the variable as the most important parameter in 100 RF iterative modeling.

**Table 1 molecules-25-01228-t001:** Tentative characterization of chemical constituents of two species of GRR by UHPLC-LTQ-Orbitrap/MS in negative ion mode.

No	Name	Formula	*m/z*	Adduction	Fragment Ion	RT	Scabra	Rigescens
**Iridoid**								
1	Secologanic acid	C_16_H_22_O_10_	373.11	[M-H]	179/108/204/282	8.40		+
2	Longanic acid *	C_16_H_24_O_10_	375.13	[M-H]	371/179	2.35	+	
3	Gentiolactone	C_10_H_12_O_5_	211.10	[M-H]	167	7.50	+	+
4	Gentiopicroside*	C_16_H_20_O_9_	401.11	[M-H+HCOOH]	179/355/149/119	5.22	+	+
5	Sweroside *	C_16_H_22_O_9_	403.12	[M-H+HCOOH]	357/195/179/125	5.63	+	+
6	8-epikingside	C_17_H_24_O_11_	403.12	[M-H]	371/223/179	7.49	+	
7	Caryptoside	C_17_H_26_O_11_	405.14	[M-H+HCOOH]	179	2.26	+	+
8	Swertiamarin *	C_16_H_22_O_10_	419.12	[M-H+HCOOH]	179/355/211/119	3.85	+	+
9	Loganin *	C_17_H_26_O_10_	435.22	[M-H+HCOOH]	389.22	7.17		+
10	Secoxyloganin	C_17_H_24_O_11_	449.13	[M-H+HCOOH]	179/241/359/403	2.66	+	
11	Morroniside	C_17_H_26_O_11_	451.14	[M-H+HCOOH]	405/243/179/	2.94	+	
12	2′-*O*-(2,3-dihyrben)-gentiopicroside	C_20_H_28_O_14_	491.14	[M-H]	167/323/459	4.59	+	+
13	2′-*O*-(2,3-dihyben)-swertamairn	C_23_H_26_O_13_	509.13	[M-H]	153/297/315/367	10.53		+
14	3′-*O*-(2,3-dihyben)-swertamairn	C_23_H_26_O_13_	509.22	[M-H]	153/517/411/321	6.11		+
15	Deglu-noneacetylate-rindoiside	C_23_H_26_O_13_	509.22	[M-H]	153/297/315/367	9.53		+
16	Rigenolide A	C_25_H_28_O_12_	519.15	[M-H]	307	11.20		+
17	4-glu+D97RT	C_26_H_34_O_11_	521.20	[M-H]	359/329	9.44		+
18	Lacriciresinol	C_26_H_34_O_11_	521.20	[M-H]	473/355/375/415	15.26		+
19	6′-*O*-d-glu swertiamarin	C_25_H_28_O_13_	535.14	[M-H]	409/491/153/339	14.39	+	
20	6′-*O*-d-glu-Loganic acid	C_22_H_34_O_15_	537.18	[M-H]	213	2.05	+	+
21	Dideacetylate-deglu-rindoside	C_25_H_28_O_14_	551.14	[M-H]	491/409/509	12.34		+
22	Gentianaside	C_22_H_36_O_13_	553.13	[M-H+HCOOH]	507	5.03	+	
23	6-*O*-d-glu-gentiopicroside	C_22_H_30_O_14_	563.16	[M-H+HCOOH]	341/517	5.69	+	+
24	Tortoside B	C_28_H_38_O_13_	581.15	[M-H]	401/357/313/269	1.47		+
25	Gentiabavaroside	C_26_H30O_15_	581.16	[M-H]	401/357/313/221	1.47	+	
26	6-*O*-d-glu-swertianmarin	C_22_H_32_O_15_	581.17	[M-H+HCOOH]	341/535/517/179	2.83	+	
27	Deacetylate-deglu-rindoside	C_27_H_30_O_15_	593.15	[M-H]	451/531/551	19.54	+	+
28	Deglu-trifloroside	C_29_H_32_O_15_	619.17	[M-H]	577	24.5	+	+
29	Deglu-gelidoside	C_29_H_32_O_16_	635.16	[M-H]	551/451/593	23.19	+	+
30	Gentiotrifloroside	C_29_H_36_O_17_	655.19	[M-H]	315/493/529	7.41	+	+
31	Deglu-scabraside	C_34_H_34_O_15_	681.18	[M-H]	639/475/153	28.45	+	+
32	Gentrigeoside A	C_36_H_60_O_12_	683.40	[M-H]	640/622	28.46	+	
33	2,3-deacetyl-trifloroside	C_32_H_40_O_17_	695.18	[M-H]	315	13.78	+	
34	6′-*O*-ace-3-*O*-glu-2-hy-sweroside	C_31_H_38_O_18_	697.17	[M-H]	315/535/571/315	14.25	+	
35	Trideacetylate-trifloroside	C_31_H_38_O_18_	697.18	[M-H+HCOOH]	505/651/313/269	8.78		+
36	Scabran G3	C_28_H_40_O_19_	725.21	[M-H]	341/383/503/679	2.55	+	+
37	2-deaceyl-trifloroside	C_33_H_38_O_19_	737.19	[M-H]	315/575/693	16.1	+	
38	Dedihydroxybenzoate-Macrophylloside	C_33_H_40_O_19_	739.20	[M-H]	697/577/613/535	15.76	+	+
39	Deacetylate-Trifloroside	C_33_H_40_O_19_	739.20	[M-H]	697/577/613/535	14.7	+	+
40	Deacetylate-Rindoside	C_33_H_40_O_20_	755.20	[M-H]	593/315/713	13.92	+	+
41	Trifloroside *	C_35_H_42_O_20_	781.22	[M-H]	619/739/577/315	18.44	+	
42	Dideacetylate-Macophylloside	C_36_H_40_O_20_	791.24	[M-H]	521/629/315	15.11	+	
43	Rindoside *	C_35_H_42_O_21_	797.21	[M-H]	315/493/635/755	17.19	+	+
44	Deacetylate-scabraside	C_38_H_42_O_19_	801.22	[M-H]	639/597	18.41	+	
45	Acetylate-trifloroside	C_37_H_44_O_21_	823.23	[M-H]	619/577/781	21.59	+	
46	Deacetylatemacrophylloside A	C_38_H_42_O_21_	833.21	[M-H]	671/697/535/315	15.33	+	+
47	Scabraside	C_40_H_44_O_20_	843.24	[M-H]	681/639/315/801	21.66	+	+
48	Dideacetylate-4′-glu-trifloroside	C_37_H_48_O_23_	859.23	[M-H]	697	19.31	+	+
49	Macrophylloside A	C_40_H_44_O_22_	875.22	[M-H]	739/577/535	18,74	+	+
50	Deacetylate-4′-glu-trifloroside	C_39_H_50_O_24_	901.26	[M-H]	577/459/535/859	13.36	+	
51	4-*O*-β-d-trifloroside	C_41_H_52_O_25_	943.27	[M-H]	459/619/901/577	16.29	+	+
52	Acetylate-4′-glu-scabraside	C_46_H_54_O_25_	1005.29	[M-H]	963/681/639/477	20.73	+	
53	4″-*O*-β-d-glucopyranosy1-6′-*O*-(4-*O*-β-d-glu-caffeoyl) linearroside	C_46_H_56_O_25_	1007.30	[M-H]	845/801/487/639	11.59	+	
54	Benzoxy-4″-*O*-β-d-glucopyranosy1-6′-*O*-(4-*O*-β-d-glucopyranosylcaffeoyl)linearroside	C_53_H_60_O_26_	1111.33	[M-H]	845/487/639/801	15.03	+	
**Flavonoids**								
55	Isovitexin	C_21_H_20_O_10_	431.10	[M-H]		9.7	+	+
56	Isoorientin(Homoorientin) *	C_21_H_20_O_11_	447.15	[M-H]	327/357/429	8.29	+	+
57	Isoscoparin	C_22_H_22_O_11_	507.17	[M-H+COOH]	461	6.66		+
58	2-glu-isovitexin	C_27_H_30_O_15_	593.15	[M-H]	551/451/531/409	19.54	+	
59	Isosaponarin	C_27_H_30_O_16_	593.15	[M-H]	311/431/473/503	7.68		+
60	Saponarin	C_27_H_30_O_17_	593.15	[M-H]	367	11.51		+
61	4′-glu-isoorientin	C_27_H_30_O_16_	609.14	[M-H]	447	5.27		+
62	Lutonarin	C_27_H_30_O_16_	609.14	[M-H]	447/519/489/327	6.87		+
63	Rutin	C_27_H_30_O_16_	609.14	[M-H]	447/519/489/327	6.9		+
64	Keampferol	C_15_H_10_O_6_	331.04	[M-H+HCOOH]	285/165	22.50	+	
65	7-glu-isopyrenine	C_29_H_34_O_17_	653.17	[M-H]	315	10.06	+	+
66	Hyperoside	C_21_H_20_O_12_	509.22	[M-H+HCOOH]	463	8.95	+	
67	Lonicerin	C_27_H_30_O_15_	593.19	[M-H]	551/451	17.06	+	
**Miscellaneous Compounds**								
68	Caffeic acid	C_9_H_8_O_4_	179.03	[M-H]	135/109	9.25	+	+
69	Ferulic acid	C_10_H_10_O_4_	193.02	[M-H]	149	3.77	+	
70	Isoferulic acid	C_10_H_10_O_4_	193.02	[M-H]	149	4.01	+	
71	Vanilloloside	C_13_H_16_O_9_	315.07	[M-H]	153	1.6	+	+
72	Glu-2,3-dihydroxybenzoic acid	C_14_H_20_O_8_	315.11	[M-H]	187/297/253/145	22.81	+	
73	Glu-2-hydro-3-methoben	C_14_H_18_O_9_	329.09	[M-H]	167	6.89	+	+
74	Methyl-3-(β-d-glucopyranosyl)-2-hydroxybenzoate	C_14_H_18_O_9_	329.23	[M-H]	209/311	19.25	+	
75	Glu-caffeic acid	C_15_H_18_O_9_	341.09	[M-H]	179/135/203/239	1.97		+
76	Syringin	C_17_H_24_O_9_	371.10	[M-H]	249	4.7	+	
77	3-[(6-*O*-Arabinopyranosyl)-β-d-glucopyranosyloxy]oct-1-en	C_19_H_34_O_10_	421.20	[M-H]	289/133	12.57	+	
78	Methyl-3-[(6-*O*-β-d-glucopyranosyl)-β-d- glucopyranosyloxy]-2-hydroxybenzoate	C_23_H_24_O_12_	491.12	[M-H]	153/315/475	12.31		+
79	(+) Syringaresinol	C_24_H_30_O_8_	491.14	[M-H+HCOOH]	315/447/153	9.94		+
80	Lonicerin	C_27_H_30_O_15_	593.19	[M-H]	551/451/	16.68		+
**Xanthones**								
81	Mangiferin *	C_19_H_18_O_11_	421.08	[M-H]	403/331/301	5.98	+	
82	Gentianabavaroside	C_26_H_30_O_15_	581.16	[M-H]	401/357/313/	3.71	+	
**Triterpenoids**								
83	Gentrigeoside D	C_36_H_60_O_13_	745.40	[M-H+HCOOH]	699	11.99		+
84	Gentrigeoside C	C_42_H_70_O_17_	845.24	[M-H]	683/803	21.77	+	
85	Glu-Gentigeoside C	C_46_H_56_O_25_	1007.31	[M-H]	845/801/639/487	11.52	+	
86	Gentrigeoside A (Dammarane)	C_36_H_60_O_12_	683.40	[M-H]	640/622	28.46	+	
87	Rha-Gentrigeoside	C_46_H_54_O_27_	1037.51	[M-H+HCOOH]	991	13.6	+	

Annotation: + indicated the compound in the specie; * indicated compound compared with standards; blank indicated without the compound.

**Table 2 molecules-25-01228-t002:** Identification of chemical markers from two species of GRR.

No.	Name	Formula	RT	*m/z*	Adduction	Fragment Ion
1	(+)Syringaresinol	C_24_H_30_O_8_	9.94	491.14	[M-H+HCOOH]	315/447/153
2	Lutonarin	C_27_H_30_O_16_	6.87	609.14	[M-H]	447/519/489/327
3	Trifloroside	C_35_H_42_O_20_	18.44	781.22	[M-H]	619/739/577/315
4	4-β-d-glu-trifloroside	C_41_H_52_O_25_	16.29	943.27	[M-H]	459/619/901/577
5	4″-*O*-β-d-glucopyranosy1-6′-*O*-(4-*O*-β-d-glucopyranosylcaffeoyl)linearroside	C_46_H_56_O_25_	11.59	1007.30	[M-H]	845/323/487/639/801
6	Macrophylloside A	C_40_H_44_O_22_	18.74	875.22	[M-H]	739/577/535
7	Scabraside	C_40_H_44_O_20_	21.66	843.24	[M-H]	681/639/315/801
8	Unknown		25.12	717.46		
9	Unknown		15.11	1105.32		
10	Unknown		19.00	891.22		
11	Unknown		15.63	1053.27		

**Table 3 molecules-25-01228-t003:** Sample collection information of GRR.

Species	location	Sample Number	Species	Species	Sample Number
*Gentiana scabra*	QY-wdz	1–20	*Gentiana scabra*	Ys-j	52
*Gentiana scabra*	QY--yem	21–39	*Gentiana rigescens*	Yx-hsc	53–57
*Gentiana scabra*	QY-	40–49	*Gentiana rigescens*	Yx-jfc	58–62
*Gentiana scabra*	Ys-y	50	*Gentiana rigescens*	Yx-xh	63–72
*Gentiana scabra*	Ys-a	51	*Gentiana rigescens*	Yx-th	73–86

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
