# Peer review of "Metabolomics Combined with Multivariate Statistical Analysis for Screening of Chemical Markers between Gentiana scabra and Gentiana rigescens"

_molecules, 2020, doi:10.3390/molecules25051228_

Round 1

Reviewer 1 Report

The study is interesting. However, data analysis must be improved. The main points to be addressed are the following.

1- Biomarker discovery must be performed both by univariate and by multivariate data analysis. Authors must add the univarite part. I suggest to use t-test or Mann-Whitney test depending on the data distribution with false discovery rate correction. Moreover, ROC analysis should be applied to the selected markers.

2- Outlier detection should be performed (for example by PCA on each single group). Please, perform outlier detection.

3- Multivariate data analysis is not performed in the right way. I agree with the use of OPLS-DA, but 1+6 latent variables is not reliable. In reliable models, the number of latent variables should be 1 or 2 in the case of 2-class problem. 7 latent variables indicates the presence of large structured noise. The results of the permutation test on the class response must be reported, the cross-validation procedure must be repeated with a different number of groups (for example 5,6,7,8) and the error in cross-validation reported. The risk is to overfit the data.

4- VIP>1 is not a right rule for biomarker discovery (see Franceschi P et al J. Chemometr. 2012;26:16-24). I suggest to use selectivity ratio (Rajalahti T et al Anal. Chem. 2009;81:2581-2590).

5- RF is an interesting approach since it is scale invariant. However, RF must be applied considering all the variable, not a subset. In the study RF is used to select markers after OPLS-DA because an efficient variable selection rule was not applied, but RF can be applied to the whole data set. I suggest to use or RF or OPLS-DA. If RF is used, the out-of-bag error must be reported.

6- The authors should address the issue of the season effect on the metabolic composition. Are the markers independent on the season?

7- The samples of the two species have been collected in different regions, specifically GS from Qingyuan and GR from Yunnan. Are there effects of the region on the metabolic composition? Region and species are confounding effects!

8- An external validation set is not included in the experimental design. Please, discuss this weakness of the study.

Minor points:

1-why is it necessary to use metabolomics to distinguish GS from GR? Is it difficult using standard methods? Please, add some comments in the introduction.

2- why were not both the data sets obtained in negative and positive ionization mode investigated? Usually, both the data sets are investigated.

3- what are the 80% and 15% rules used in data pre-processing?

Author Response

Reviewers' comments:

Question 1: Biomarker discovery must be performed both by univariate and by multivariate data analysis. Authors must add the univariate part. I suggest to use t-test or Mann-Whitney test depending on the data distribution with false discovery rate correction. Moreover, ROC analysis should be applied to the selected markers.

Response: Thank you for your valuable suggestions, univariate data analysis has been added as Section 2.3.1 in the revised manuscript. In Section 2.3.1, 283 features with P-value < 0.05 were screened out, and the heat map illustrated the differences in 283 features between the two Gentiana species was shown in Fig. 8. Furthermore, considering that the ROC analysis results of all 283 differential metabolites were too much, and could not be well presented in the revised manuscript. Therefore, ROC analysis of 11 markers selected by multivariate data analysis was added in the Section 2.3.3.

Question 2: Outlier detection should be performed (for example by PCA on each single group). Please, perform outlier detection.

Response: Thank you for the suggestions. In the study, we did not perform outlier detection in multivariate analysis for the following reasons: (1) The metabolic profiles of herbal materials were varied across batches, reflecting the adaptability of plants to different growth conditions. Therefore, it was normal that the metabolic profiles of different batch samples were different. Although some batches of samples deviated from the majority of data distribution of the two categories, i.e., 95% confidence intervals of PCA analysis. We could not hastily identify these samples as outliers, on the base of small sample sizes in this study. (2) In order to get a comprehensive understanding of the chemical difference between two Gentiana species, it was necessary to cover the chemical variation of a given species as far as possible, so as to improve the usability and generality of the results of multivariate analysis. For the above reasons, we believe that outlier detection was not suitable for this study.

Question 3: Multivariate data analysis is not performed in the right way. I agree with the use of OPLS-DA, but 1+6 latent variables is not reliable. In reliable models, the number of latent variables should be 1 or 2 in the case of 2-class problem. 7 latent variables indicates the presence of large structured noise. The results of the permutation test on the class response must be reported, the cross-validation procedure must be repeated with a different number of groups (for example 5,6,7,8) and the error in cross-validation reported. The risk is to overfit the data.

Response: Thank you for the valuable suggestions. The permutation tests (n = 200) of two categories of OPLS-DA model were performed to validate the model performance. As shown in Fig. 10b, the values of R2 = (0.0, 0.394) and Q2 = (0.0, -0.664) of category 1 and R2 = (0.0, 0.385) and Q2 = (0.0, -0.672) of category 2 indicated that the OPLS-DA model in our study without the risk of overfitting. The results of this part have been added in the revised manuscript (Section 2.3.3.).

Question 4: VIP>1 is not a right rule for biomarker discovery (see Franceschi P et al J. Chemometr.2012;26:16-24). I suggest to use selectivity ratio (Rajalahti T et al Anal. Chem. 2009;81:2581-2590).

Response: We agreed with your opinions that VIP>1 was not a right rule for biomarker discovery. In this study, the biomarker discovery was divided into two steps, the rule of VIP>1 in OPLS-DA for search differential metabolites was the first step, which was only a rough screening to reduce the calculations of RF iterative modeling.

Question 5: RF is an interesting approach since it is scale invariant. However, RF must be applied considering all the variable, not a subset. In the study RF is used to select markers after OPLS-DA because an efficient variable selection rule was not applied, but RF can be applied to the whole data set. I suggest to use or RF or OPLS-DA. If RF is used, the out-of-bag error must be reported.

Response: Your valuable suggestions are greatly appreciated. Because random parameter affected the analysis results of RF model, RF was modeled through 100 iterations to get consistency results. Although this approach yielded reliable results, the computational overhead was enormous. Therefore, in order to minimize the computational workload, OPLS-DA model was used for the rough screening in the first step, and the iterative modeling of RF was used for the detailed analysis. The out-of-bag error rate of RF model established by 11 selected metabolic characteristics was added in the revised manuscript (Section 2.3.3.).

Question 6: The authors should address the issue of the season effect on the metabolic composition. Are the markers independent on the season?

Response: We thank you for the valuable advice. The 86 batches of GRR samples were collected in the official harvest season of each species stipulated by Chinese Pharmacopoeia at their main production sites, whether the observed markers were affected by season effect need to be further explored. In the further research, we will continue to collect experimental samples from different seasons to elucidate the metabolic composition with season effect.

Question 7: The samples of the two species have been collected in different regions, specifically GS from Qingyuan and GR from Yunnan. Are there effects of the region on the metabolic composition? Region and species are confounding effects!

Response: We agreed with the reviewer’s comment. Region and species mainly affected the metabolic composition of TCM. In this study, we mainly focused on screening the chemical markers between GS and GR for authentication. In order to meet the representativeness of samples, the two species were collected from representative main production areas. In the next step, we will explore the metabolic composition variations of GS and GR caused by region or species difference.

Question 8: An external validation set is not included in the experimental design. Please, discuss this weakness of the study.

Response: We sincerely appreciate the reviewer’s comment. Indeed, an external validation set was an important part in the experimental design, we will collect more samples for the experimental validation. The weakness of the study on experimental design has been discussed in the Conclusion section.

Minor points:

Question 1: why is it necessary to use metabolomics to distinguish GS from GR? Is it difficult using standard methods? Please, add some comments in the introduction.

Response: Thank you for reviewer’s constructive comments. The standard methods (morphological identification, thin-layer identification, microscopic discrimination, barcode identification) could be used to distinguish GS from GR, but morphological identification and microscopic discrimination existed subjective error by researcher, thin-layer identification and barcode identification were only used to distinguish GS from GR without any chemical constituents and chemical markers information. Metabolomics may be a better choice for authentication of GS from GR, it could provide more useful chemical information. The introduction has been revised and replenished carefully. Please see the Introduction section.

Question 2: why were not both the data sets obtained in negative and positive ionization mode investigated? Usually, both the data sets are investigated.

Response: We appreciate reviewer’s suggestion. According to the typical Base Peak ion (BPI) of GRR in both negative and positive ionization mode in Figure 2, more useful information was prevailed in the negative ionization mode. Therefore the data sets were only obtained in negative ionization mode.

Question 3: what are the 80% and 15% rules used in data pre-processing?

Response: For the data filtering. Firstly, features with relative standard deviation (RSD) greater than 15% in QC samples were removed, because these features were unstable during the data collection of whole experiment. Secondly, the remaining features were filtered according to the 80% rule, where features that presented in at least 80% of samples in one group were allowed to remain.

Reviewer 2 Report

The paper “Metabolomics combined with multivariate statistical analysis for screening of chemical markers between Gentiana scabra Bge. and Gentiana rigescens Franch” submitted by Dean Guo and coworkers, describes the identification and characterization of chemical components of the two previously mentioned plant species using UHPLC coupled with LTQ-Orbitrap. OPLS-DA and Random Forest iterative modeling were used to differentiate the two species.

Although the central subject of this paper is well justified and represented an interesting metabolomic approach to differentiate the two species and to establish chemical markers of Gentianae Radix et Rhizoma from different sources, is necessary to make extensive modification to the manuscript before being considered to published in Molecules. The following points must be addressed.

1.- Since the authors include an abbreviation section, is important to include all the necessary ones to understand the paper, for example, in the row 81 the abbreviation “QC” (Quality Control??) is used, but its meaning is not enlisted in the abbreviation section.

2.- The Table 2, although informative about the characteristic of the different column chromatography tested is absolutely unnecessary. In any case this table could be moved to a Supplementary Material file.

3.- In the overall it is very difficult to read the paper due to the deficient use of the English language. Is strongly recommended the revision of the manuscript by an English native spoken person.

4.- The Figure 8 is missing in the manuscript.

5.- The mass spectrometry fragmentation patterns of the iridoids, flavonoids and xanthones must be supported by adequate references and not only in a deductive way. For these classes of natural products there is a lot of information regarding fragmentation patterns under mass spectrometry conditions.

Author Response

Question 1.- Since the authors include an abbreviation section, is important to include all the necessary ones to understand the paper, for example, in the row 81 the abbreviation “QC” (Quality Control??) is used,

Response: We sincerely appreciate reviewer’s constructive comments. In the manuscript, the abbreviation “QC” was used for Quality Control exactly. Moreover, we have checked and revised the Abbreviation section carefully. Please see this section.

Question 2.- The Table 2, although informative about the characteristic of the different column chromatography tested is absolutely unnecessary. In any case this table could be moved to a Supplementary Material file.

Response: Thank you very much for your suggestion. We have moved the Table 1 about the detailed information of the different screened separation columns to the Supplementary Materials. Please see the Supplementary Material file.

Question 3.- In the overall it is very difficult to read the paper due to the deficient use of the English language. Is strongly recommended the revision of the manuscript by an English native spoken person.

Response: Thanks for your suggestion. We have solicited an English native speaker to have polished our manuscript.

Question 4.- The Figure 8 is missing in the manuscript.

Response: We are sorry for the confusion. In fact, the Figure 8 was in the page 15 of the manuscript.

Question 5.- The mass spectrometry fragmentation patterns of the iridoids, flavonoids and xanthones must be supported by adequate references and not only in a deductive way. For these classes of natural products there is a lot of information regarding fragmentation patterns under mass spectrometry conditions.

Response: Thanks very much for your kind advices. The mass spectrometry fragmentation patterns of the iridoids flavonoids and xanthones has been summarized according to the references [1-3] and data collected by HUPLC-LTQ-Orbitrap/MS, the references were listed as follows:

  1. Dinda, B., Pharmacology and Applications of Naturally Occurring iridoids. Springer: Switzerland, 2019.
  2. Suying, L.; Fengrui, S.; Zhiqiang, L., Mass spectrometry analysis of traditional Chinese medicine. Science Press Beijing, 2012.
  3. QIN Jie-ping, D. J.-g., FENG Yu-qi, FENG Xu, Applications of Organic Mass Spectrometry in Structure Identification of an Impurity Compound in Prepared Mangiferin Extracted from Mangifera indica L. leaves. Journal of Chinese Mass Spectrometry Society 2008, 29.

Round 2

Reviewer 2 Report

I deeply thank the authors for having taken my recommendations into account. With the changes made, I consider that the work meets the scientific merits to be published in Molecules